# Implications of the three-dimensional chromatin organization for genome evolution in a fungal plant pathogen

David E. Torres[1,2,5], H. Martin Kramer[1,5], Vittorio Tracanna [3,5], Gabriel L. Fiorin [1], David E. Cook [1,4], Michael F. Seidl [1,2,6] & Bart P. H. J. Thomma [1,3,6]

The spatial organization of eukaryotic genomes is linked to their biological functions, although it is not clear how this impacts the overall evolution of a genome. Here, we uncover the three-dimensional (3D) genome organization of the phytopathogen *Verticillium dahliae*, known to possess distinct genomic regions, designated adaptive genomic regions (AGRs), enriched in transposable elements and genes that mediate host infection. Short-range DNA interactions form clear topologically associating domains (TADs) with gene-rich boundaries that show reduced levels of gene expression and reduced genomic variation. Intriguingly, TADs are less clearly insulated in AGRs than in the core genome. At a global scale, the genome contains bipartite long-range interactions, particularly enriched for AGRs and more generally containing segmental duplications. Notably, the patterns observed for *V. dahliae* are also present in other *Verticillium* species. Thus, our analysis links 3D genome organization to evolutionary features conserved throughout the *Verticillium* genus.

The spatial organization of eukaryotic genomes is directly linked to their biological functions, although underlying mechanisms remain largely unclear. Many plant pathogenic fungi display a distinct genome organization, commonly termed the "two-speed genome", in which gene-poor and repeat-rich genomic regions contain genes that mediate host infection, display increased plasticity, and typically display features of heterochromatin[1,2]. These dynamic compartments are paramount for the coevolutionary 'arms-race' between pathogens and their hosts, potentially enabling the avoidance of host immune recognition and the evolution of novel functions in pathogenicity[2–5].

In eukaryotic nuclei, physically separated genomic sites colocalize, while proximal sites may be separated through folding barriers into a three-dimensional (3D) genome structure that comprises various levels of organization[6,7]. Local 3D interactions shape chromosome structure into discrete genomic regions, commonly known as topologically associating domains (TADs); self-interacting genomic regions that are delineated by boundaries that display less physical interaction[8]. Although their function is still controversial, TADs have been associated with transcriptional regulation by governing the impact of regulatory sequences on nearby genes[8–10]. Other studies implicate TADs in genome replication by synchronizing origins of replication[11]. Interestingly, various studies point towards the evolutionary conservation of TAD organization among related organisms[12–14]. While the genome clearly contains different chromatin states and 3D genome organization, their exact roles in genome function and evolution remain unclear.

In the fungi *Saccharomyces cerevisiae*, *Schizosaccharomyces pombe*, and *Neurospora crassa*, the 3D genome organization is linked

[1]Laboratory of Phytopathology, Wageningen University and Research, Droevendaalsesteeg 1, 6708 PB Wageningen, The Netherlands. [2]Theoretical Biology & Bioinformatics Group, Department of Biology, Utrecht University, Utrecht, The Netherlands. [3]University of Cologne, Institute for Plant Sciences, Cluster of Excellence on Plant Sciences (CEPLAS), Cologne, Germany. [4]Department of Plant Pathology, Kansas State University, 1712 Claflin Road, Manhattan, KS, USA. [5]These authors contributed equally: David E. Torres, H. Martin Kramer, Vittorio Tracanna. [6]These authors jointly supervised this work: Michael F. Seidl, Bart P.H.J. Thomma. ✉e-mail: m.f.seidl@uu.nl; bthomma@uni-koeln.de

to heterochromatin distribution[15–17]. Determination of the 3D genome of the endophytic fungus *Epichloë festucae* has revealed that heterochromatic, repeat-rich regions frequently colocalize with TAD boundaries and can be implicated in genome folding[18]. Moreover, in planta-induced genes are enriched near those regions[18]. These findings suggest an intimate link between the 3D genome, dynamic genomic compartments, heterochromatin, and conditional gene expression.

The asexual soil-borne fungal plant pathogen *Verticillium dahliae* is a notorious broad host-range vascular wilt pathogen[19]. The *V. dahliae* genome is characterized by the occurrence of extensive large-scale genomic rearrangements that are associated with segmental duplications that underwent substantial reciprocal gene losses and are enriched in active transposable elements (TEs)[20–23]. This organization results in distinct dynamic genomic compartments, previously termed lineage-specific regions due to the abundant presence–absence variations, and presently referred to as adaptive genomic regions (AGRs)[20–22,24]. These AGRs display unique chromatin characteristics such as the enrichment of H3K27me3, depletion of 5mC methylation, and high chromatin accessibility, and are enriched for in planta-induced genes and other conditionally responsive genes that contribute to environmental adaptation[20,23–25]. However, it presently remains unclear whether and how the 3D genome affects the organization and evolution of the *V. dahliae* genome. Here, we explore the chromatin conformation of *V. dahliae* with DNA proximity ligation followed by sequencing (Hi-C) to uncover the spatial organization of the AGRs within the genome.

## Results

### Topologically associating domains in the *Verticillium dahliae* genome

To uncover the 3D organization in *V. dahliae* strain JR2, we performed Hi-C in two highly correlated biological replicates (Supplementary Fig. S1a–d), revealing that interaction strength generally negatively correlated with genomic distance (Supplementary Fig. S1c). To query for the occurrence of TADs, the genome was divided into bins (average bin size ~4 kb) and insulation scores were calculated along the genome based on physical interaction strengths between adjacent bins (Supplementary Fig. S1g–i). These insulation scores reflect how isolated a genomic region is from its surroundings. Given that a TAD is a self-interacting genomic region with sequences that physically interact more with each other than with sequences outside the TAD, bins that display a significant local dip in their insulation score, and are thus depleted in physical interactions with neighboring bins, were consequently identified as a TAD boundary region that separate TADs (Fig. 1b). In other words, as TADs are characterized by strong local interactions, TAD boundaries can be identified as regions that are depleted in such interactions in-between TAD regions that show strong interactions. The lower the insulation score, the stronger the boundary, and thus the more distinct TADs occur in the DNA structure, while higher scores imply more interaction between neighboring regions, and possibly less structured TADs. Using this approach, we identified 353 TADs (mean size 102 kb) separated by 345 boundaries (mean size 4.7 kb, excluding the telomeric ends) along the eight *V. dahliae* chromosomes (Fig. 1a, b, d and Supplementary Figs. S1e, f and S2). While 277 TADs (78.47%) and 308 boundaries (88.76%) localizes. in the core genome, 76 TADs (21.53%) and 39 boundaries (11.24%) localized in AGRs. Interestingly, the insulation of TADs is weaker in AGRs than in the core genome (Fig. 1c).

As observed previously[23–27], the core genome is generally enriched in H3K27ac and H3K4me2, while centromeres and TE-rich core regions are enriched in H3K9me3 and DNA methylation, and AGRs are enriched in H3K27me3 (Fig. 1d, e and Supplementary Fig. S3). Such broad chromatin associations are maintained similarly on TADs and boundaries in core and AGRs (Supplementary Fig. S4a, b), suggesting that chromatin characteristics mainly associate with the overall genome compartmentalization rather than with TAD organization.

Nevertheless, we also observe that TADs and boundary regions in core and AGRs differ in chromatin accessibility, histone modifications, and DNA methylation (Fig. 1e and Supplementary Fig. S4b), and these observations suggest that TADs and boundaries differ in functionality, not only between each other, but also between the two compartments.

Given the little information on chromatin and TAD organization in filamentous fungi, we further assessed TAD predictions at different binning thresholds (Supplementary Fig. S2a), while maintaining that an accurate TAD prediction should accurately depict the previously experimentally verified centromeric TADs[27]. This analysis revealed that binning at 4 kb resolution most accurately called centromeric TADs (Supplementary Fig. S2c). TAD prediction based on binning at 2 kb resolution that recapitulates smaller TADs (mean size 51.5 kb) and loops resulted in an increased number of TADs and TAD boundary regions (539) as nested TADs were identified when compared with predictions at 4 kb resolution (Supplementary Fig. S2). In total, 493 TAD boundary regions overlap with the core genome, 46 with AGRs and five with centromeres. Importantly, even at this 2 kb resolution, TAD boundary regions in the core genome and in AGRs maintain the features that we identified based on the 4 kb binning resolution (Supplementary Fig. S2), as we observe that TADs and boundary regions differ in gene density, repeat density, chromatin accessibility, H3K27me3 and sequence conservation (Supplementary Fig. S2d). In conclusion, the most robust TAD prediction based on validation based on centromeric TADs occurs at 4 kb binning resolution. However, even if we capture smaller TAD sizes, and thus reveal nested TADs while permitting centromeric TADs to be predicted sub-optimally, we maintain the differential hallmarks of TADs and boundary regions that suggesting that TADs and boundaries differ in functionality between the two genomic compartments.

### Local TAD organization associates with transcriptional regulation

Given the chromatin differences between TADs and boundaries in AGRs and the enrichment of AGRs in conditionally responsive genes[20,24,25], we hypothesized that transcriptional profiles of core genes and those in AGRs differ between TADs and boundaries. Interestingly, boundary regions in the core genome are significantly enriched in genes and consequently depleted of TEs when compared with TADs (Fig. 2a). However, within AGRs we did not observe differential enrichment of genes or TEs in TADs versus boundaries (Fig. 2a). To assess the impact of TAD organization on gene expression, we integrated genomic, transcriptomic, and chromatin data for all *V. dahliae* genes, and performed uniform manifold approximation and projection (UMAP) for dimensional reduction[24,25]. As observed previously, genes separated based on these characteristics show core and AGR groupings (Supplementary Fig. S4c)[24]. In addition, when considering only boundary genes (Supplementary Fig. S4c), genes in AGRs are enriched for H3K27me3, while core genes are enriched for H3K27ac. Furthermore, there was a clear separation based on transcriptional activity, with genes in core genome boundaries generally displaying higher transcription levels, associated with increased H3K4me2 levels, than genes in AGR boundaries (Supplementary Fig. S4c). Interestingly, boundary insulation and in vitro expression level positively correlated (Fig. 2b and Supplementary Fig. S4c), indicating that genes in weakly insulated TAD boundaries are generally lower expressed than genes in strongly insulated boundaries. Overall, genes located within TAD boundaries are lower expressed than those in TADs, both for the core genome and AGRs (Fig. 2c), a trend we similarly observed when we assayed gene expression of *V. dahliae* during infection of the thale cress *Arabidopsis thaliana* (Supplementary Fig. S5a). However, the expression of genes within TAD boundaries in the core genome is notably higher than the expression of genes within boundaries in AGRs, whereas genes further away from boundaries in the core genome and AGRs are similarly expressed (Fig. 2c).

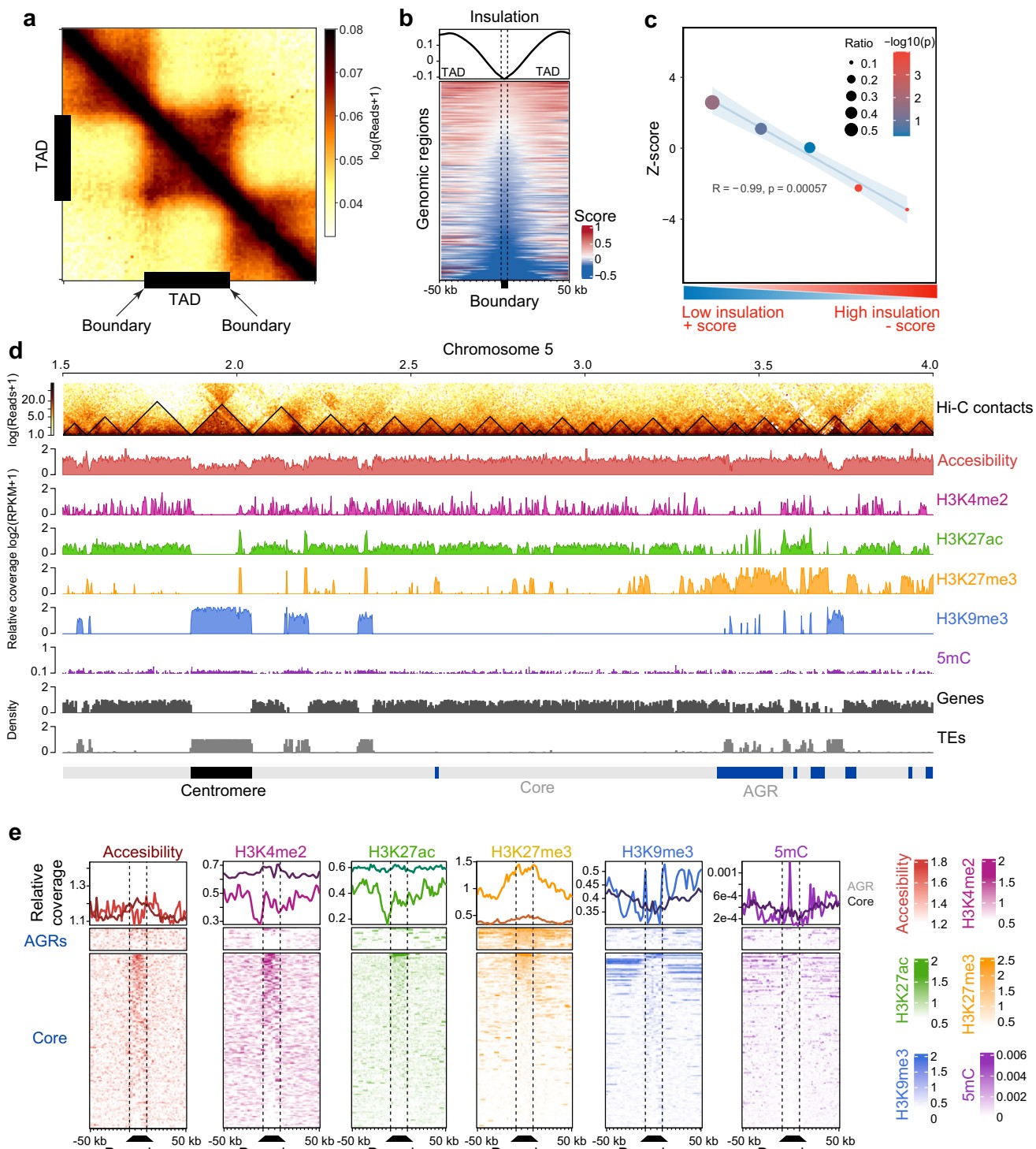

**Fig. 1 | The *Verticillium dahliae* genome is organized in topological associating domains (TADs). a** Hi-C contact matrix showing local interaction frequency, aggregated over predicted TADs (black bars) with 50 kb up- and downstream sequence. The drop in intensity at boundaries at either edge of the TADs indicates stronger interaction within TADs than with neighboring genomic regions. **b** Heatmap showing the physical interaction strength (insulation score) centered over boundaries with 50 kb up- and downstream sequence as rows, ordered on insulation score with weakest insulated boundaries on top. The top plot displays the average of the insulation scores in the heatmap below. **c** TADs in AGRs are weaker insulated when compared with core genome TADs. The *X* axis indicates quintiles of boundaries, separated based on insulation scores. The *Y* axis indicates *z*-score and the −log10(*P* value) color-scale after a one-sided permutation test for enrichment of boundaries in AGRs (10,000 iterations). The plot displays a linear

regression (blue line) and confidence interval (light blue) as well as the R and *P* value for the linear regression. **d** TAD distribution in *V. dahliae* strain JR2, exemplified by a section of chromosome 5. From top to bottom: Hi-C contact matrix depicting TADs as black triangles, open chromatin determined with ATAC-seq, histone modifications H3K4me2, H3K27ac, H3K27me3, and H3K9me3 normalized over a micrococcal nuclease digestion control, GC methylation, as well as gene and transposable element (TE) densities in 10 kb windows. Adaptive genomic regions (AGRs) and the centromeric region are indicated in blue and black, respectively. **e** Chromatin characteristics are differentially associated with TAD boundaries in the core genome and in AGRs. On top, the distribution of each chromatin feature centered for boundaries (dashed lines) with 50 kb up- and downstream sequence, for the core genome (dark line) and AGRs (light line). On the bottom, the corresponding heatmaps are shown. Source data are provided as a Source Data file 1.

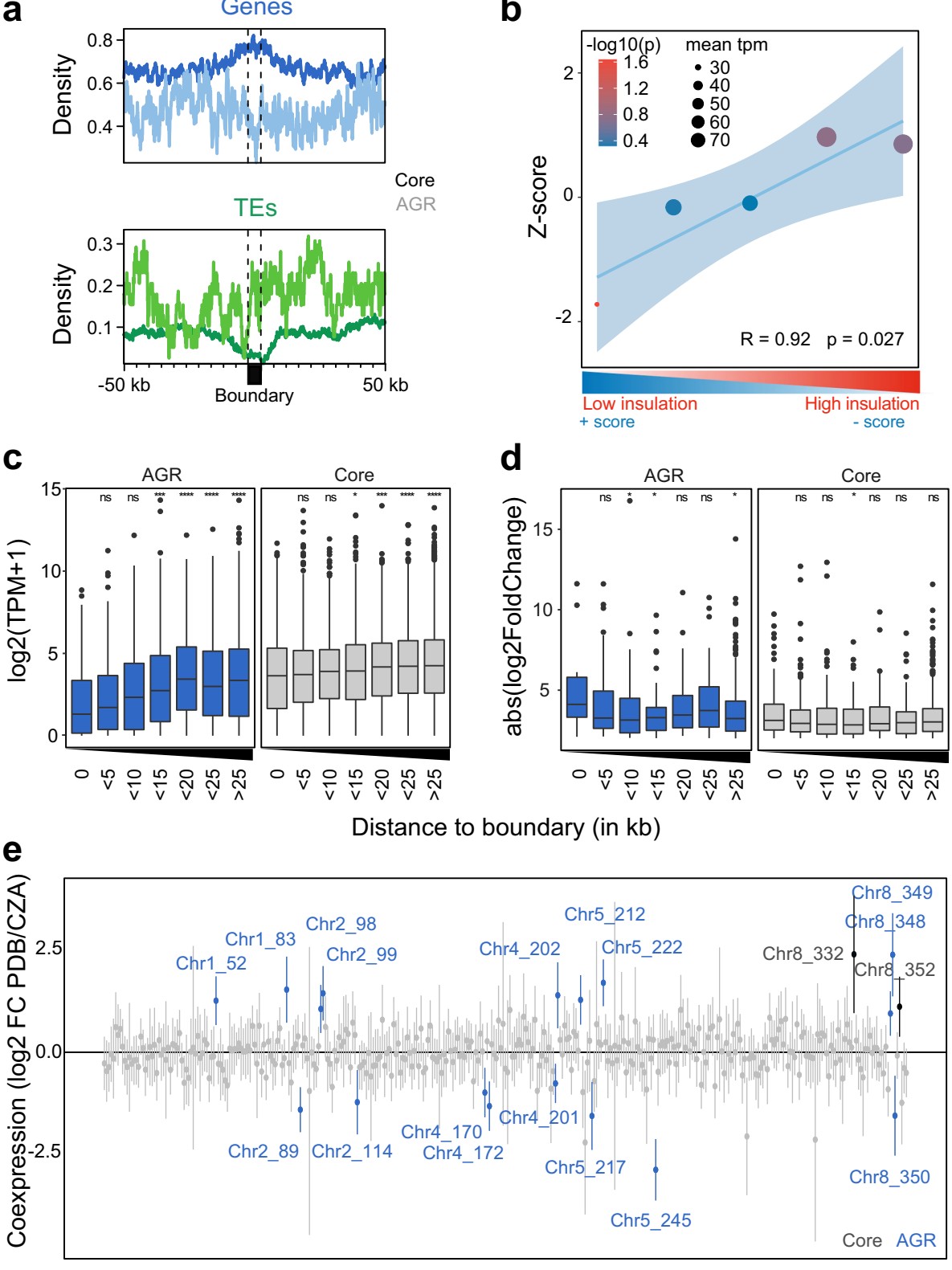

As TADs may govern differential gene expression, we hypothesized that differentially expressed genes (DEGs) in vitro and upon plant infection are enriched within TADs and depleted in boundaries. We previously showed that DEGs are enriched in AGRs and in H3K27me3-rich regions in the core genome[20,24,25], indicating that TADs may function as regulatory units both in the core genome and in AGRs. We observed that genes at distances of 5–15 kb and >25 kb from TAD boundaries in AGRs are significantly weaker differentially expressed than genes located in AGR boundaries (Fig. 2d). In planta differentially expressed genes are only significantly stronger differentially expressed at distances <5 kb from TAD boundaries (Supplementary Fig. S5b), thus collectively suggesting that differential gene

**Fig. 2 | TAD organization is associated with transcription in *Verticillium dahliae*. a** Average density of genes and transposable elements (TEs) per 1 kb window centered over boundaries with 50 kb up- and downstream sequence in the core genome and in adaptive genomic regions (AGRs) of *V. dahliae* strain JR2. The core genome (dark line) and in AGRs (light line). **b** Genes in lowly insulated boundaries are lower expressed than those in more highly insulated boundaries. Association between TAD boundary quintiles, separated on insulation score, and transcription of genes located in TAD boundaries. The *Y* axis depicts the *z*-score, color of the datapoints indicates the −log10(*P* value) after a one-sided permutation test (10,000 iterations) and size of datapoints indicates mean transcription value (TPM) of represented genes. A linear regression (blue line), with 95% confidence interval (light blue), between boundary quintiles is displayed. **c** Transcription values for *V. dahliae* cultivated for 6 days in potato dextrose broth (PDB) and **d** absolute log2-fold change in expression between cultivation in PDB or in Czapec-Dox medium (CZA), for all genes grouped based on their distance to the closest boundary in the core genome (gray) or in AGRs (blue). Statistically significant differences in average transcription level for the distance groups was compared to the group of genes located in boundaries (distance 0) and determined by the one-sided Wilcoxon rank-sum test (\*P<=0.05, \*\*P < =0.01, \*\*P < 0.001, \*\*\*\*P < =0.0001). Black lines in the boxes depict the median, boxes extend from first to third quartile, vertical lines indicate the 1.5× interquartile range and dots depict outliers for each category. Normalized mean of biologically independent experiments n = 3 is shown. **e** Linear regression effect size of each TAD on differential gene expression between cultivation for 6 days in PDB or in CZA. Mean effect size of each TAD is shown as a point, with 95% confidence interval, and TADs with a significant effect (95% confidence interval is significantly different from 0) are shown in color and labeled by corresponding chromosome and TAD number, for TADs in the core genome (black labels) and in AGRs (blue labels). Normalized mean of biologically independent experiments *n* = 3 is shown. Source data are provided as a Source Data file 2.

expression in vitro and in planta in core and in AGRs is generally not enhanced at TADs.

To investigate whether genes localizing within the same TAD in *V. dahliae* display transcriptional co-regulation, we fitted a linear model in which differential expression of each gene in vitro is predicted by TAD association. We identified 19 TADs with a significant effect on co-transcription (Fig. 2e). Of these, 17 are associated with AGRs and only two with the core genome (Fig. 2e), suggesting that transcriptional co-regulation of expression mainly occurs in AGRs. In total, 68 out of 258 (26.4%) TADs in the core genome, while 64 out of 96 (66.7%) TADs in AGRs contain more than five DEGs (Supplementary Fig. S4d). Of these TADs, 13 out of 68 (19.1%) and 19 out 64 (29.7%) in the core genome and AGRs, respectively, contain more than twice the number of genes that display co-regulation of differential transcription (Supplementary Fig. S4d). We similarly identified 50 TADs with a significant effect on co-transcription of *V. dahliae* genes during *A. thaliana* infection (Fig. S5c). Due to low fungal biomass most genes are lower expressed in planta compared with in vitro condition, yet if we only consider co-regulated TADs with increased expression in planta, nearly all TADs (18 out of 24) localize in AGRs. Thus, our results imply that TAD organization transcription in vitro and in planta and that, although some TADs display transcriptional co-regulation of gene expression, this occurs only for a subset of TADs that predominantly locate in AGRs.

## TAD boundaries are conserved
To study TAD conservation within *V. dahliae*, we analyzed Hi-C data of *V. dahliae* strain VdLs17 that is 98% syntenic to strain JR2 (Fig. 3a)[21], and predicted 365 TADs (mean size 99 kb) and 357 boundaries (mean size 4.5 kb) (Supplementary Fig. S6a–c). Notably, the TAD organization in VdLs17 displays similar patterns of insulation scores, gene enrichment, and TE-depletion as in JR2, suggesting that TAD characteristics are conserved in *V. dahliae* (Supplementary Fig. S6c–h). Moreover, based on the distribution of TAD boundaries over syntenic regions between VdLs17 and JR2 (*n* = 342 and *n* = 330 TADs, respectively), we observed a significant overlap and correlation of insulation scores between boundaries of the two strains (*n* = 225, *P* = 9.6 × 10$^{-4}$, one-way Fisher exact test; Fig. 3a and Supplementary Fig. S6g), and an overall overlap in TAD positions (Fig. 3a, b). Also, we observe that non-syntenic regions in VdLs17 are enriched for weak TAD boundaries (*z*-score = 2.3858, *P* = 0.00001, permutation test after 10,000 iterations; Supplementary Fig. S6h), a characteristic of AGRs in *V. dahliae* strain JR2.

Genomic comparisons between *V. dahliae* strains have revealed extensive genomic rearrangements and structural variations (SVs)[20–23]. However, as TAD boundaries are conserved between *V. dahliae* strains JR2 and VdLs17, we hypothesized that boundaries may lack such genomic variation. We used genome sequencing data of 42 *V. dahliae* strains[23,24] to query the occurrence of single nucleotide variants (SNVs) and presence/absence polymorphisms (PAVs) over TAD boundaries in *V. dahliae* strain JR2. Indeed, deletions, duplications, inversions, and translocations occur more commonly in TADs than in boundaries, indicating depletion of genomic variation from boundaries (Fig. 3c, d). One possibility is that genomic variation at boundaries negatively impacts *V. dahliae* (i.e., purifying selection). To assess this, we calculated the expected amount of SV breakpoints and SV coverage occurring in boundaries and found that SVs occur less frequently than expected in boundaries localized in syntenic regions (Fig. 3e). Interestingly, while we observed a depletion of SNVs and of PAVs in boundaries in the core genome (Fig. 3c, d), boundaries in AGRs showed increased PAV (Supplementary Fig. S7c) combined with lower nucleotide diversity (Supplementary Fig. S7a, b). Moreover, SVs occur more commonly over boundaries in non-syntenic regions and in AGRs, which agrees with previous observations that SVs occur frequently in non-syntenic regions (Fig. 3e)[23]. Collectively, these findings suggest that TAD boundaries in the core genome are strongly conserved, while boundaries in AGRs are evolutionary less stable.

In *V. dahliae*, SVs often colocalize with polymorphic TEs that display PAV between 42 strains and are evolutionary young, scarcely methylated, and highly expressed[23]. As TE activity may mediate the formation of SVs[22,23], we investigated whether polymorphic TEs occur more frequently in TADs than in boundaries. We identified 36 polymorphic TEs (21.8% of the total) that display PAV between *V. dahliae* strains JR2 and VdLs17. However, we observed no overrepresentation nor depletion of polymorphic TEs in boundary regions (Fig. 3e). Nevertheless, interestingly, some TE insertions in *V. dahliae* strain JR2 occur at sites of boundary differences (Fig. 3f and Supplementary Fig. S7c, d), suggesting that polymorphic TEs lead to changes in TAD organization.

## Conservation of TAD organization throughout the *Verticillium* genus
Given the conservation of TAD organization among *V. dahliae* strains, we investigated conservation throughout the *Verticillium* genus based on synteny to *V. dahliae* strain JR2 (Supplementary Fig. S7e) and calculated nucleotide conservation scores[28]. As expected, we observed higher conservation scores for the core genome than for AGRs (Fig. 4a). Moreover, like *V. dahliae* (Fig. 3c–e and Supplementary Fig. S7f), we observed increased conservation scores for boundaries versus TADs in the core genome, but not in AGRs (Fig. 4a), indicating that core boundaries are conserved within the *Verticillium* genus.

To assess conservation of TAD organization, we compared the sequence and position of all TAD boundaries in *V. dahliae* strain JR2 to the genome sequences of the other *Verticillium* species (Supplementary Fig. S8a). We used previously generated Hi-C data[27] to assess whether the insulation score of boundaries predicted based on sequence and position is lower than for adjacent genomic regions, i.e., TADs (Supplementary Fig. S8a). Employing this strategy to the genome of *V. dahliae* VdLs17, we recovered 269 boundaries (Supplementary Fig. S8b–d) that display significant positional overlap with the

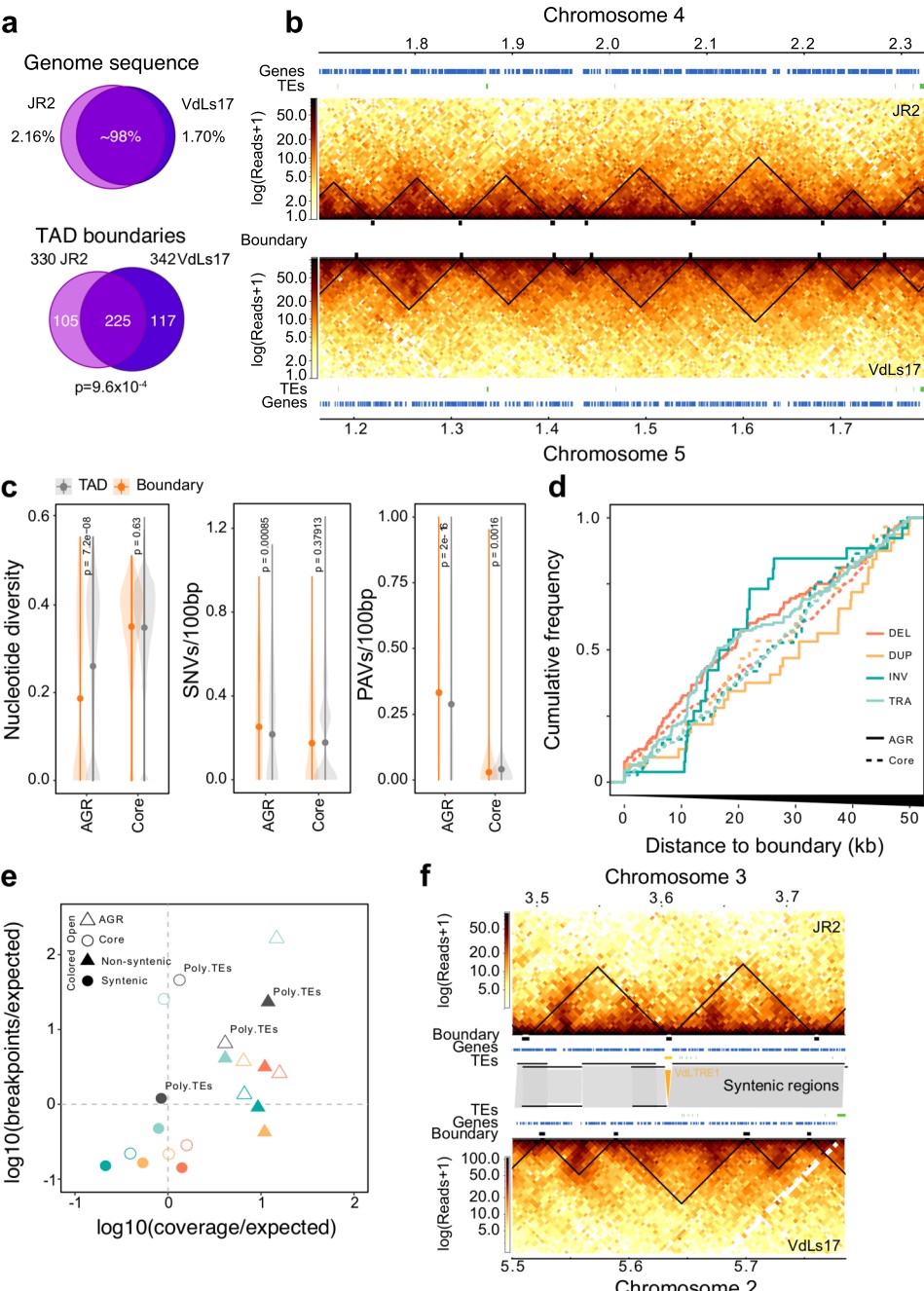

**Fig. 3 | Topological associating domain (TAD) organization is conserved in *Verticillium dahliae*. a** Top: *V. dahliae* strains JR2 and VdLs17 are highly similar as 97.84% and 98.30% of their respective genomes are syntenic. Bottom: Most of the TAD boundaries overlap between JR2 and VdLs17. *P* value after one-way Fisher's exact test. **b** Syntenic block between JR2 chromosome 4 and VdLs17 chromosome 5 shows conserved distribution of TADs and boundaries. Heatmaps represent contact matrixes of JR2 (top) and VdLs17 (bottom) with TADs (black triangles). Genes and transposable elements (TEs) are displayed above and below. **c** Boundaries are not enriched for genomic variation in a set of 42 *V. dahliae* strains. Nucleotide diversity, single nucleotide variants (SNVs), presence/absence variation (PAVs). TADs *n* = 277, 76 for core genome and AGRs, respectively. Boundaries *n* = 308, 39 for core genome and AGRs, respectively. *P* values based on a one-way Wilcoxon rank-sum test. Center dot depict the median; line ranges depict the upper and lower 1.5× interquartile range. **d** Cumulative frequency plot of structural variant

(SV) breakpoints over distance from boundaries in the core genome (dashed line) and in AGRs (solid line), overlaps with boundaries (distance = 0) are included. SVs are separated in deletions (DEL, orange), duplications (DUP, yellow), inversions (INV, green) and translocations (TRA, blue)[23]. **e** TAD boundaries in AGRs and in the core genome contain more and fewer SVs than expected by chance, respectively. SVs in boundaries in the core genome (open circles) and in AGRs (open triangles) are indicated, as well as in boundaries in syntenic (solid circles) and non-syntenic (solid triangles) genomic regions and in polymorphic TEs (gray circles). **f** Synteny breaks associated with transposable elements affect TAD organization. Heatmaps represent contact matrixes of JR2 (top) and VdLs17 (bottom) with TADs (black triangles), and TADs, genes and TEs are displayed in-between. Syntenic regions are indicated as gray blocks. A VdLTRE1 insertion in strain JR2 is indicated in yellow. Source data are provided as a Source Data file 3.

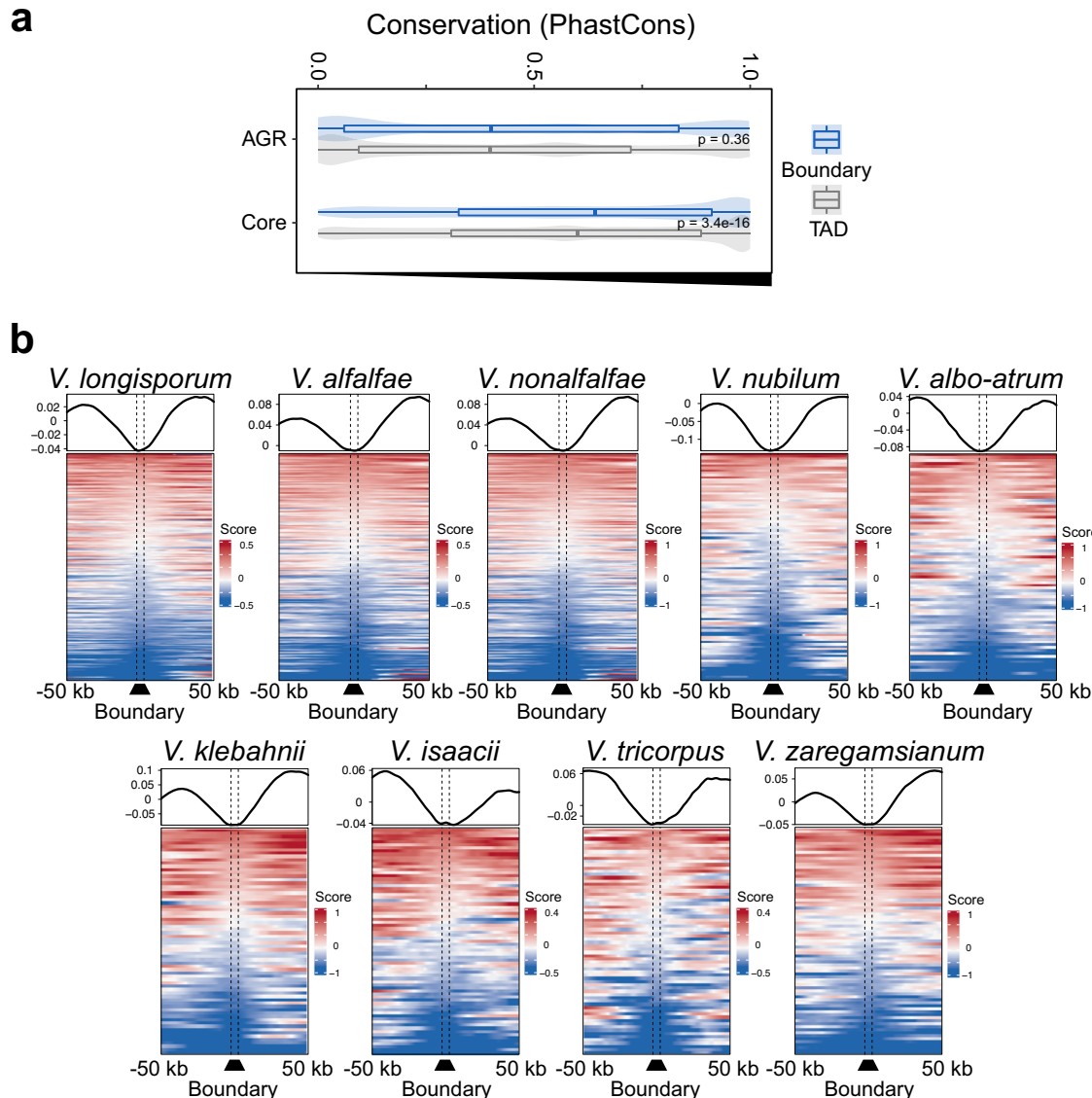

**Fig. 4 | TAD boundaries show signs of conservation in the *Verticillium* genus.**
**a** TAD boundaries are more conserved than TADs. Boxplots display the conservation score of each TAD (gray) and boundary (blue) in the core genome and in AGRs. TADs $n = 277, 76$ for core genome and AGRs, respectively. Boundaries $n = 308, 39$ for core genome and AGRs, respectively. $P$ values based on a one-way Wilcoxon rank-sum test. Boxplots depict center line, median; box limits, upper and lower quartiles; whiskers, 1.5× interquartile range. **b** Insulation score over TAD boundaries predicted by the sequence-based method, with 50 kb up- and downstream sequence, for each *Verticillium* species. Line plots display average signals over boundaries and up/downstream sequence, bottom plots display predicted boundaries in rows, ordered by insulation scores for each independent experiment. Source data are provided as a Source Data file 4.

boundaries as determined with the insulation method ($z$-score = 27.1264, $P = 9.99 \times 10^{-5}$; Supplementary Fig. S8b–e). Thus, we next used this approach for the other *Verticillium* spp. as well. In general, the sequence-predicted boundaries in the *Verticillium* genus depict a drop in insulation score with the adjacent genomic regions (Fig. 4b), indicating that we correctly assigned TAD boundaries. The results show that boundaries of *V. dahliae* strain JR2 are more likely to be shared with closer phylogenetic species (Supplementary Fig. S8f). For instance, only 80 boundaries were recovered in the more distantly related *V. albo-atrum*, whereas 254 boundaries were predicted in *V. alfalfae* and 283 boundaries in *V. longisporum*, both close relatives of *V. dahliae* (Supplementary Fig. S8f). In addition, we recovered full-length TAD structures in syntenic regions of the other *Verticillium* species, indicating high TAD conservation in the core genome (Supplementary Fig. S8f). Collectively, our results suggest that TADs and their boundaries are conserved among *Verticillium* species consistent with phylogenetic distance.

## Adaptive genomic regions physically colocalize

Besides local genome architecture, we assessed physical associations between distal genomic regions. Making use of previously demonstrated centromeric interactions as references[27], we identified 889 additional genomic regions that consistently colocalize. Interestingly, of these, 475 (53.4%) involve AGRs (Fig. 5a, b and Supplementary Table S1), which is a strong overrepresentation (chi-squared test; $P < 0.05$, Supplementary Table S2) given that AGRs only represent 3.33 Mb (9.6%) of the 36.15 Mb total genome size[24]. Moreover, colocalization events among AGR regions comprise nearly one-third of the noncentromeric interactions (28.1%), whereas AGR-core and core–core colocalization events comprise 25.3% and 46.6%, respectively. As expected, TADs with a significant effect on co-transcription in AGRs (Fig. 2e and Supplementary Fig. S5c) physically colocalize in vitro and upon plant colonization ($z$-score = 1.40, $P = 0.0098$, $z$-score = 2.59, $P = 0.0196$; permutation test in vitro and in planta, respectively),

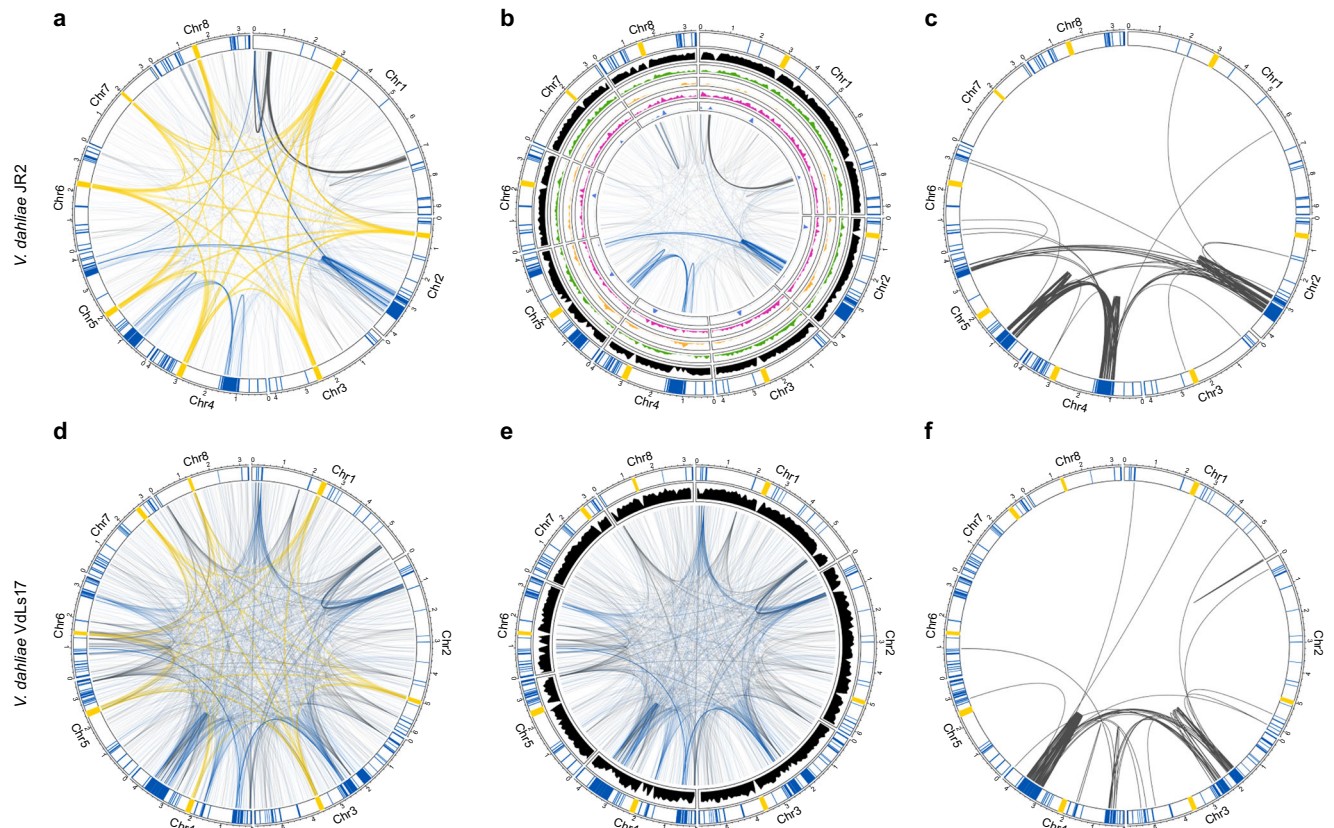

**Fig. 5 | Adaptive genomic regions physically colocalize in *Verticillium dahliae*.** All circular plots display in the outer track the eight chromosomes of *V. dahliae* with centromeres highlighted in yellow, adaptive genomic regions (AGRs) in blue, and core regions in white. Long-range interactions in *V. dahliae* strain JR2 (**a**) and VdLs17 (**d**) that exceed the average interaction strength of centromeres are shown as edges. Edges for centromeric interactions are shown in yellow, AGR interactions in blue, and core interactions in gray. Noncentromeric long-range interactions in *V. dahliae* strains JR2 (**b**) and VdLs17 (**e**) are shown as edges and the gene density (10 kb) is shown in black. For *V. dahliae* strain JR2, the inner tracks depicting histone modification densities. From outside to inside: H3K27ac (green), H3K27me3 (yellow), H3K4me2 (purple), and H3K9m3 (blue). **c, f** Edges represent segmental duplications. Source data are provided as a Source Data file 5.

suggesting that transcriptional co-regulation of expression mainly occurs in close proximity to AGRs.

The colocalization of centromeres in *V. dahliae* strain JR2 correlates with CENH3 nucleosomes, *VdLTRE9* (LTR/Gypsy), and H3K9me3[27]. However, in our attempts to identify epigenetic drivers of long-range interactions, we found no correlation between colocalizing AGR regions and any of the histone modifications H3K9me3, H3K27me3, H3K4me2, and H3K27ac, nor chromatin accessibility (Fig. 5b). Given that all AGRs are involved in individual bipartite colocalization events, and while AGRs share characteristics they are sequence diverse, it may not be surprising that long-ranger interactions do not have a simple epigenome association (Fig. 5b). We previously showed that *V. dahliae* evolution involved large-scale segmental duplications[22,23]. Intriguingly, the colocalizing AGR regions are associated with duplication events (Fig. 5c); of the 475 colocalization events that involve AGR regions, 260 involve segmental duplications (Fig. 5c, Supplementary Fig. S9, and Supplementary Table S3), which is a significant enrichment not only genome-wide (z-test, $P < 0.05$), but even within AGRs (z-test, $P < 0.05$). Moreover, whereas genome-wide 264 interactions were recorded that involve segmental duplications, nearly all (260; 98.5%) concern AGRs.

To assess the conservation of AGR colocalization in *V. dahliae*, we similarly analyzed Hi-C data of the VdLs17 strain. Interestingly, also in VdLs17, noncentromeric colocalization events are enriched for AGR interactions that involve 452 out of 1451 (31.2%) interactions (chi-squared test; $P = 8.5 \times 10^{-56}$; Fig. 5d, e). Moreover, we similarly observed that interacting AGRs are strongly associated with segmental

duplications (Fig. 5f, Supplementary Fig. S9, and Supplementary Table S1). Together, these results indicate that long-range interactions between AGRs represent a conserved genome organization in *V. dahliae*.

We finally assessed whether the long-range colocalization patterns are conserved throughout the *Verticillium* genus (Fig. 6 and Supplementary Tables S3 and S4). Interestingly, for several *Verticillium* species, we could demonstrate that AGRs are overrepresented in long-range interactions (Supplementary Table S2), namely in *V. albo-atrum*, *V. klebahnii*, *V. nubilum*, and *V. nonalfalfae*. Interestingly, also in these species, segmental duplications are enriched in long-range interactions. Moreover, although we were not able to demonstrate enrichment for interactions between AGRs in *V. tricorpus*, *V. alfalfae*, *V. isaacii*, and *V. longisporum*, long-range interactions predominantly occurred among segmental duplications in all species except *V. alfalfae* (Supplementary Fig. S11). Thus, long-range interactions are conserved across the *Verticillium* genus and involve segmental duplications that are likely instrumental for AGR formation.

## Discussion
Based on Hi-C analyses we uncovered the local and global 3D genome organization of *V. dahliae* and related species. The *V. dahliae* genome contains clear TADs that display reduced insulation in AGRs, as well as significantly enhanced co-regulation of gene expression compared with the core genome. Notably, TADs are conserved in the *Verticillium* genus and their boundaries generally lack structural variation.

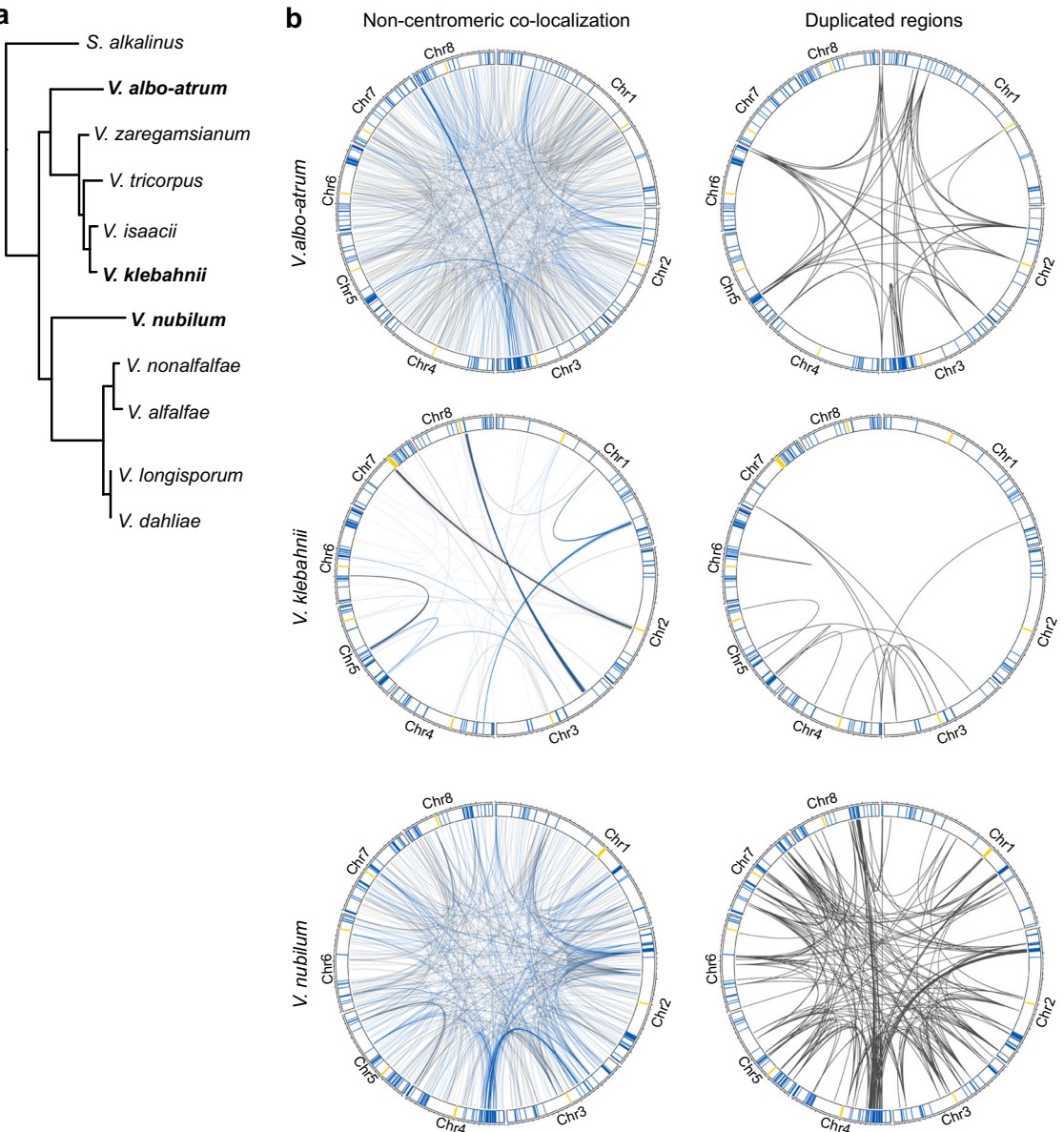

**Fig. 6 | Adaptive genomic regions physically colocalize across the**
***Verticillium* genus. a** Taxonomic relationship between ten *Verticillium* species.
**b** Noncentromeric interactions occur preferentially between adaptive genomic
regions (AGRs) in different *Verticillium* species. For all circular plots, outer track

depicts centromeres in yellow, AGR regions in blue, and core regions in white. For
every genome, noncentromeric interactions that exceed the average interaction
strength of centromeres are shown. The right circular plots display segmental
duplications for each genome. Source data are provided as a Source Data file 6.

Intriguingly, we show that AGRs physically interact throughout the
*Verticillium* genus, albeit not in an all-versus-all interaction like cen-
tromeres, but rather pairwise. These interactions are associated with
segmental duplications that help to define AGRs. Collectively, our
findings link 3D genome organization with genome function and
evolution throughout the *Verticillium* genus.

TADs have been described for many eukaryotes, including
fungi[11,15–18]. *V. dahliae* TADs are smaller (-100 kb) than typical mam-
malian TADs (-0.2–2.5 Mb)[8], yet similar to those in *Drosophila* and
other fungi[11,16,18,29]. In metazoans, TAD boundaries are bound by CTCF
(CCCTC-binding factor) proteins that act as insulator elements for
cohesin-mediated loop extrusion, and are considered hallmarks of
TADs[30,31]. Given that CTCF proteins have not been identified in fila-
mentous fungi[32], annotation of fungal analogs as TADs may be con-
sidered controversial. However, TAD-like domains also occur in
bacteria, known as chromosomal interaction domains (CIDs), and

recently the global transcription repressor Rok was identified as a
functional analog of the mammalian CTCF insulator elements[33]. A
similar situation may be true for fungi, where additionally many other
features are shared with metazoan TADs.

As DNA interacts more strongly within than between TADs, they
separate genomes into discrete units that, in several organisms,
coordinate DNA replication[11,34,35]. In addition, TADs have been impli-
cated in transcriptional co-regulation[18,36–38], although a causal role in
coordination of gene expression remains controversial[9]. Similar to
studies on the fungi *Rhizophagus irregularis* and *E. festucae*[18,38], we find
that only few *V. dahliae* TADs display transcriptional co-regulation.
Nonetheless, it is still interesting that for this smaller subset of TADs,
they all reside in AGRs that are epigenetically distinct from the core
genome, involving lack of TE methylation, enrichment in H3K27me3,
and accessible DNA[24]. Although in *V. dahliae*, similar to other plant-
associated fungi[39–42], H3K27me3 plays a role in transcriptional

regulation, we also revealed that this modification does not act as the global regulator of differential gene expression at AGRs[25], and additional factors must be involved in transcriptional regulation.

We note that our TAD predictions, based on validation based on centromeric TADs, may encompass nested and smaller TADs. We cross-verified the main features associated with our TAD prediction at 4 kb binning resolution also based on a binning at 2 kb resolution, which may better recapitulate shorter and nested TADs (mean TAD size of 51.5 kb; Supplementary Fig. S2). Even though predictions of centromeric TADs are clearly compromised at this binning resolution (Supplementary Fig. S2), importantly, TAD boundary regions in the core genome and in AGRs maintain the main differential features that were identified at 4 kb binning resolution, as they differ in gene density, repeat density, chromatin accessibility, H3K27me3 and sequence conservation. Thus, we believe that we have not only performed robust TAD predictions, but also identified thorough characteristics that can differentially characterize TADs in the core genome and in AGRs, likely underpinning their differential functionalities.

TAD boundaries are typically conserved between close relatives[12,13,43–45], which we similarly observe in *Verticillium*. However, in contrast to the core genome, TAD boundaries in AGRs are less conserved, implying that TAD organization in the evolutionary younger and dynamic AGRs still needs to settle or is less strictly defined. It is interesting to note that we observed TE insertions near newly generated, or extensively rearranged, TAD boundaries in AGRs. Cultivar-specific TAD boundaries in cotton were reported to generally harbor more TEs than TAD boundaries that are shared between cultivars[46]. Moreover, de novo TE insertions in humans generated new TAD boundaries[47]. TE activity is largely confined to AGRs in *V. dahliae*[22,23], and may thus be involved in modulating TAD organization. How this impacts longer-term trajectories of these regions remains to be determined.

Besides local interactions in TADs, the 3D genome displays long-distance interactions within and between chromosomes. We previously reported centromere clustering in *V. dahliae*[27], and now revealed long-range interactions among AGRs. In *N. crassa* such interactions occur between heterochromatic regions, e.g., between H3K27me3 domains[16,48]. Thus, we hypothesized that H3K27me3-marked AGRs similarly colocalized in *V. dahliae*. However, as not all AGRs interact, despite their enrichment in H3K27me3, and AGRs interact in a pairwise fashion following the pattern of segmental duplications, we conclude that we currently cannot assign a single epigenetic mark as a clear driver for colocalization. It is tempting to speculate that physical clustering of AGRs in the nucleus involves membrane-less nuclear bodies and liquid–liquid phase separation that allows spatial segregation of e.g., transcription and DNA-repair[49,50]. In addition, association with nuclear membranes could cause spatial segregation of AGRs. In mammalian nuclei, heterochromatic regions are associated with lamin proteins and additional anchor proteins to form lamina-associated domains (LAD) at the nuclear membrane[51,52]. However, LAD proteins have not been found in fungi[53], and more generally it is currently not clear what might drive these specific long-range DNA interactions.

The 3D organization of chromosomes influences timing of replication[35,54,55] and may lead to differential replication timing of AGRs and core genome. Such differences in replication timing have been observed for H3K27me3-rich regions and have been associated with chromosome instability in *Z. tritici*[41,56,57]. Such instability, and physical colocalization of highly homologous sequences, may lead to decreased DNA separation efficiency during mitosis and increased DNA double-strand breaks that may explain the increase in genomic rearrangements in AGRs[20,24,58].

Altogether, we have uncovered a novel phenomenon that contributes to the divergence of plastic regions and the core genome, by showing that their 3D organization differs with clear impact on evolution and transcriptional regulation. This holistic view that combines detailed genetic, epigenetic, and spatial information fosters further understanding of genome function and evolution in fungi.

## Methods

### Hi-C analysis and TAD prediction

Hi-C library preparation was performed with *V. dahliae* strains JR2 and VdLs17 as previously described[27], and paired-end (2 × 150 bp) sequenced on the NextSeq500 platform at USEQ (Utrecht, the Netherlands). Additional Hi-C datasets of *V. dahliae* strains JR2 and VdLs17, *V. albo-atrum* strain PD747, *V. alfalfa* strain PD683, *V. isaacii* strain PD618, *V. klebahnii* strain PD401, *V. longisporum* strains PD589 and VLB2, *V. nonalfalfae* strain T2, *V. nubilum* strain 397, *V. tricorpus* strain PD593, and *V. zaregamsianum* strain PD739 were previously generated[27,59].

For each Hi-C dataset, sequenced read-pairs were quality-filtered and trimmed using Trimmomatic (v 0.36) in paired-end mode with default settings[60]. Filtered and trimmed reads were mapped to the corresponding genomes[21,27] using Burrows–Wheeler aligner (BWA mem, settings: -A1 -B4 -E50 -L0)[61]. Hi-C interaction matrices were built and analyzed using HiCExplorer tools[62]. First, we used hicBuildMatrix to generate the interaction matrix based on the in silico *Dpn*II restriction digested corresponding genome. To determine the optimal bin resolution, we used the TAD prediction at centromeric regions that we previously experimentally determined by immunoprecipitation of CenH3 and repeat content[27]. Matrix resolution was reduced by merging five adjacent bins using hicMergeMatrixBins. TAD predictions were compared to results from additional tools and hicExplorer was chosen based on its superior performance on predicting centromeric TADs. For *V. dahliae* strains JR2 and VdLs17, replicates were corrected separately according to the iterative correction and eigenvector decomposition (ICE) method[63] using hicCorrectMatrix, and TADs were predicted using hicFindTADs (settings: --delta 0.01). The tool hicFindTADs calculates the insulation score based in the number of interactions between adjacent bins, and predicts a TAD boundary region if a significant local minimum in physical interactions is observed between two adjacent bins. This local minimum result in negative values when a large difference in physical interactions between two adjacent bins occurs, thus resulting in a "strong" insulation between TADs. A local minimum in physical interactions can still be observed between adjacent bins which results in "weak" insulated TADs. Correlation between replicates was determined by using a reproducibility score based on a stratified cross-correlation using the HiCRep package[64].

To combine replicate matrices, resolutions of raw matrices were reduced by merging five adjacent bins using hicMergeMatrixBins, normalized between replicates using hicNormalize (settings: --setToZeroThreshold 1), corrected separately according to the ICE method using hicCorrectMatrix, and finally combined using hicSumMatrices[62]. For the other *Verticillium* species, matrix resolution reduction and correction was performed as above, and hicFindTADs was used to generate a table with per bin insulation scores.

### Characterization of epigenetic profiles

Chromatin immunoprecipitation followed by sequencing (ChIP-seq) for H3K4me2, H3K9me3, H3K27me3, and H3K27ac, and the assay for transposase-accessible chromatin followed by sequencing (ATAC-seq) were performed for *V. dahliae* strain JR2 as described previously[24,25,27]. ChIP datasets were normalized over MNase control samples.

We used the umap-learn implementation through the R/umap package. This implementation make use of the python UMAP algorithm[65]. For the gene analysis, the following variables were used: GC content, ATAC-seq, 5mC, H3K27ac, H3K27me3, H3K9me3 and log2(PDB in vitro expression +1), with the following parameters random_state = 42, n_neighbors = 50, n_components = 2, min_dist = 0.01,

metric = cosine. The resulting two-dimensional values from UMAP fit.transform were used for plotting and further statistical analysis using Matplolib, Numbpy and Seaborn V 0.8.1[66–68].

### Characterization of transcriptional regulation

RNA sequencing of *V. dahliae* strain JR2 cultivated for six days in potato dextrose broth (PDB) and Czapec-Dox medium (CZA) was previously performed[24,25]. RNA sequencing of *Arabidopsis thaliana* inoculated with *V. dahliae* JR2 was performed at 28 dpi as previously reported[24]. Analyses of gene[21] and TE presence[23,27] over TADs and TAD boundaries were performed using the EnrichedHeatmap package in R[69,70]. To assess the co-regulation of genes within TADs, we used R to fit a linear model with log2-fold-change in the expression of target genes between PDB and CZA as the response variable and TAD membership as a predictor, similarly as previously described[18].

### Characterization of genomic variation

Structural variants (duplications, deletions, inversions and translocations), single nucleotide variants, nucleotide diversity, presence/absence variations and polymorphic transposable elements were previously identified using paired-end sequencing reads of each 42 previously sequenced *V. dahliae* strains[23]. Briefly, structural variants were predicted using the "sv-callers" workflow with few modifications that enabled parallel execution of multiple SV callers[71], an approach that is considered optimal as it exploits complementary information to predict SVs[72,73]. Single nucleotide variants were identified using the -HaplotypeCaller of the Genome Analysis Toolkit (GATK) v.4.0[74], and the average pairwise nucleotide diversity was calculated in 1 kb sliding windows (500 bp sliding) over the genome, as previously[23]. Presence Absence Variations were identified using whole-genome alignments of DNA sequence reads from the 42 *V. dahliae* strains to the reference genome assembly of *V. dahliae* strain JR2 and summarized in 100 bp non-overlapping windows[23]. Transposable element PAV was analyzed using TEPID v.2.0[75]. To investigate if SVs and polymorphic TEs colocalize with TAD boundaries, we summarized the overlap of each set of variants by their breakpoint frequency (start or ends ±1 bp of the feature) and coverage (number of bases covered) across the genome of *V. dahliae* strain JR2[13]. Similar to Fudenberg and Pollard (2019), we calculated the log10(observed/expected) of each feature representing the deviation from a uniform distribution across the genome, therefore accounting for the proportion of the genome covered by a specific genomic feature. Finally, we considered two scenarios: core genome vs AGRs, and syntenic regions between JR2 and VdLs17 versus non-syntenic regions. Syntenic regions between *V. dahliae* strains JR2 and VdLs17 were previously determined[22]. Briefly, whole-genome alignments between the eight chromosomes were performed using MUMmer 3.0 and GEvo[76,77], where only gene-coding regions were used as anchors between syntenic chromosomal regions.

To further expand our analysis of *V. dahliae* to the full genus, we used the recently available Hi-C-corrected genomes of all *Verticillium* species[27,59]. The phylogenetic tree was generated using Realphy v. 1.12 using a maximum likelihood inferred by RAxML[78,79]. We aligned the *Verticillium* genomes using ProgressiveCactus[80]. This approach allowed us to reduce the reference-bias and consider more accurate further analysis. We obtained the specific MAF alignments on JR2 and syntenic regions using the HAL package[81]. To analyze the nucleotide conservation throughout the genus, we used PhastCons, a hidden Markov model-based method that estimates the probability that each nucleotide belongs to a conserved element based on a multiple sequence alignment guided by the established phylogenetic relationships[28]. Briefly, for each independent JR2 chromosome, we assumed a neutral evolution model and correction for indels. For further analysis, we summarized the PhastCons score over TADs and TAD boundaries in the core genome and in AGRs.

### TAD boundary prediction throughout the *Verticillium* genus

The Hi-C datasets of the *Verticillium* species (excluding *V. dahliae*), were available with one biological replicate. Therefore, we decided to predict TAD boundaries based on sequence homology to boundaries in *V. dahliae* strain JR2. We first filtered the boundary sequences that do not have a TE insertion and queried them to the *Verticillium* genomes using Blastn, retaining those with >50% coverage that were contiguous in the same syntenic block. Finally, we cross-referenced those putative TAD boundary regions with the previously calculated insulation score for each independent species.

### Statistical analysis and visualization

Hi-C matrix and TAD visualizations were performed using HiCExplorer and FAN-C[82]. Heatmap and enrichment visualization of insulation scores over boundaries, normalized chromatin marks, structural and nucleotide variants, as well as the PhastCons score, were performed using the R/EnrichedHeatmap v1.2 package[69]. Permutation tests were computed using R/Bioconductor regionR v1.18.1 package[83] and performed with 10,000 iterations, using overlaps between TAD boundaries divided by the insulation score quantiles and the predefined AGRs, and circular randomization to maintain the order and distance of the regions in the chromosomes. All statistical analyses and comparison tests were performed in R v.3.6.3[70], and visualized with ggplot2[84].

### Identification of significant colocalization events from Hi-C data

Expected–observed interaction read counts were obtained from HiCExplorer (version 3.7) through the export function with expected–observed nonzero from the corrected and summed matrices. Expected–observed counts matrixes containing the interactions between all the merged bins were filtered to keep only physical interactions between physically distant regions. Two regions were considered physically distant if the bins belonged to different chromosomes or if two bins were more than 20 kb distant from each other. Next, the bins were annotated based on which genomic compartment they belong to (centromere or AGR) and the remaining bins were annotated as core genome. Mean centromere-to-centromere expected–observed read counts is calculated and only interaction events with expected–observed counts above this threshold are kept and thus, involved in long-range interactions. It needs to be noted that our analysis is limited by the fact that we only assess those interactions that are stronger than the average interaction strength that occurs between centromeres in each species but, given that the constitution of centromeres differs between *Verticillium* spp.[27] this interaction strength may consequently not be uniform between species. Visualization of the interaction and associated genetic and epigenetic features was done in R with the circos package.

### Identification of duplicated regions and association with long-range interactions

Self-whole-genome alignments of the genome assemblies of *Verticillium* species were performed using MUMmer[76], regions that mapped elsewhere on the genome with a nucleotide identity above 80% and above 1 kb in size are considered duplicated. For each colocalization event that associate two distant regions, we verify if there is a duplicated region that borders with or overlaps with both distant regions. We consider the colocalization event to be neighboring a duplicated region, if the average distance between the colocalizing region and the duplicated region is below 50 kb.

We calculated the enrichment of colocalization events neighboring duplicated regions through a permutation test. We simulated random interactions genome-wide and calculated the number of observed interactions neighboring a duplicated region. We repeated the process 100 times to generate a distribution of expected interactions neighboring duplicated regions and ran a *t* test versus the

observed value. This permutation test was performed using the genome-wide-duplicated regions distribution and AGR compartment specific.

### Determination of adaptive genomic regions

Nucleotide sequences from the *V. dahliae* strain JR2 AGR compartment were aligned versus each of the *Verticillium* genomes included in this work using ProgressiveCactus[80]. Syntenic regions in the respective genomes are considered AGR regions in the respective organism. In addition, each *Verticillium* strain genome was aligned using ProgressiveCactus[80] with the other *Verticillium* strain genomes included in this work. Genomic regions which were unique and not found in any other genome are also considered AGR in that strain.

### Reporting summary

Further information on research design is available in the Nature Portfolio Reporting Summary linked to this article.

## Data availability

The Hi-C, ChIP-seq, RNA-seq, ATAC-seq, and Bisulfite-seq data used in this study are available in the NCBI database under the Bioproject accession codes PRJNA592220 and PRJNA641329. The genome sequence of the reference *Verticillium dahliae* isolate JR2 and VdLs17 used in this study is available at NCBI under the accession codes GCA_000400815.2 and GCA_000952015.1, respectively. Source Data is available at https://doi.org/10.5281/zenodo.10579548.

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

## Acknowledgements

This work was supported by the Consejo Nacional de Ciencia y Tecnología, México to D.E.T. (scholarship no. 2018-000009-01EXTF-00188), by a PhD grant of the Research Council Earth and Life Sciences (project 831.15.002) to H.M.K., and by Human Frontier Science Program Postdoctoral Fellowship (HFSP, LT000627/2014-L), by USDA's National Institute of Food and Agriculture (award no. 2018-67013-28492) through the Plant Biotic Interactions Program, and by the National Science Foundation (award no. 1936800) Division of Molecular and Cellular Biosciences to DEC. Work in the laboratories of M.F.S. and B.P.H.J.T. is supported by the Research Council Earth and Life Sciences (ALW) of the Netherlands Organization of Scientific Research (NWO). B.P.H.J.T. acknowledges funding by the Alexander von Humboldt Foundation in the framework of an Alexander von Humboldt Professorship endowed by the German Federal Ministry of Education and Research is furthermore supported by the Deutsche Forschungsgemeinschaft (DFG, German Research Foundation) under Germany´s Excellence Strategy – EXC 2048/1 – Project ID: 390686111.

## Author contributions

M.F.S., D.E.C., and B.P.H.J.T. conceived the project. D.E.T., H.M.K., V.T., M.F.S., and B.P.H.J.T. designed the experiments. D.E.T., H.M.K., V.T., and G.L.F. carried out the experiments, D.E.T., H.M.K., V.T., G.L.F., D.E.C., M.F.S., and B.P.H.J.T. analyzed the data. D.E.T., H.M.K., V.T., M.F.S., and B.P.H.J.T. wrote the manuscript. All authors read and approved the final manuscript.

## Funding

## Competing interests

The authors declare no competing interests.
