## [Peer Review File · Nature Communications]

Implications of the three-dimensional chromatin organization for genome evolution in a fungal plant pathogenEditorial Note: This manuscript has been previously reviewed at another journal that is not operating a transparent peer review scheme. This document only contains reviewer comments and rebuttal letters for versions considered at *Nature Communications*.

Reviewer #5 (Remarks to the Author):

In the revised manuscript 'Implications of the three-dimensional chromatin organization for genome evolution in a fungal plant pathogen' by Torres and collaborators, several points raised by reviewer #4 were addressed. However, some of them were partially addressed.

Comment 1: The authors are right and in the original article, the hicFindTADs algorithm is well explained. However, this explanation is too complex and it would be nice if the authors could make an effort to put it into simple words. Detection of TADs and boundaries is fundamental in this article, thus, it has to be understandable for a broad audience that may not be expert in Hi-C analysis.

Comment 3: With some exceptions (centromeric TADs and random regions along the chromosomes), TADs and boundaries do not seem to be well identified in the revised version. Several examples can be observed in Fig1, Fig 3, Sup Fig 2, Sup Fig6 and Sup Fig7 (in most of the cases nested TADs make difficult the visualization of the identified TADs and boundaries). This could be simple due to the heterogeneity of the Hi-C signal, but also to other patterns that are observed (and not discussed). Among these patterns, we can observe inter-TADs contacts, loops, and some patterns that may look like sliding loops (for instance in chromosome 8, in one of the AGRs). Furthermore, all figures displaying TADs need to be improved, they are small and their visualization is difficult. For instance, the insulation score is plotted below the contact map, but it is useless since the scale is small and the '0' is not defined. Furthermore, authors refer to 'a weak insulation score', but what does this mean? It would be useful that they define which score leads them to this conclusion: is it a value close to zero? Is it a value close to -1? What does '0' mean in terms of insulation? Why the score is flat along the chromosome, except in well defined TADs (for instance centromeric TADs). Despite good quality controls, the contacts displayed in the matrices do not correlate with the TADs and boundaries detected. Considering that the median size of TADs in *V. dahliae* is about 100kb and the doubts raised in the method used, it would be worth to test other tools to identify TADs and boundaries (for instance: DI (Dixon, 2012) or chromosight (Matthey-Doret C, 2020)) that are known to efficiently identify smaller TADs.

Comment 7: Given the nature of the article, it is essential to display in the main text not only a contact map for the whole genome, but also a Hi-C map centered in one individual chromosome, and one showing inter-chromosome interactions (similar to Sup Fig8d).

Concerning Figure S8, why the authors do not include the features of interest in S8C? In this figure it can be seen that in addition to the centromeric inter-chromosome contacts, there are looped regions at TADs boundaries. What are these contacts? Are they ARGs?

Why Figure S8C shows almost no contacts? For this figure, it would be nice to show the map for the whole chromosome and a magnification of the contacts so that the reader can have the 3D context of the interactions.

Other Important points:

In the abstract, the authors stated: 'Intriguingly, TADs are less clearly structured in AGRs than in the core genome...' This is a strong confirmation that needs to be shown. The authors mentioned that the insulation score is weaker in AGRs than in other areas, how is this connected to a less structured TAD? . If authors wish to state this, it would be a good idea to analyze contact dynamics within ARGs and core genome, and define after this what it is 'a clearly structured region'.

Circular plots are helpful (probably better placed in supplementary figure) but undoubtedly, matrices showing ARGs inter-contacts need to be presented, given that this is a major finding of the article.

Reviewers' Comments:

Reviewer #5 (Remarks to the Author):

In the revised manuscript 'Implications of the three-dimensional chromatin organization for genome evolution in a fungal plant pathogen' by Torres and collaborators, several points raised by reviewer #4 were addressed. However, some of them were partially addressed.

Comment 1: The authors are right and in the original article, the hicFindTADs algorithm is well explained. However, this explanation is too complex and it would be nice if the authors could make an effort to put it into simple words. Detection of TADs and boundaries is fundamental in this article, thus, it has to be understandable for a broad audience that may not be expert in Hi-C analysis.

Authors: We thank this reviewer for the appreciation of our work. We agree with the reviewer that the understanding of TADs and boundaries detection is fundamental for understanding our work. Thus, we elaborated to explain logic of using the insulation score as method to determine TAD boundaries. We now state: *“Given that a TAD is a self-interacting genomic region with sequences that physically interact more with each other than with sequences outside the TAD, bins that display a significant local dip in their insulation score, and are thus depleted in physical interactions with neighboring bins, were consequently identified as a TAD boundary region that separate TADs (Fig. 1b). In other words, as TADs are characterized by strong local interactions, TAD boundaries can be identified as regions that are depleted in such interactions in-between TAD regions that show strong interactions. The lower the insulation score, the stronger the boundary, and thus the more distinct TADs occur in the DNA structure, while higher scores imply more interaction between neighboring regions, and possibly less structured TADs.”*

In addition, we also elaborate somewhat in the Online Methods section and now state: *“The tool hicFindTADs calculates the insulation score based in the number of interactions between adjacent bins, and predicts a TAD boundary region if a significant local minimum in physical interactions is observed between two adjacent bins. This local minimum result in negative values when a large difference in physical interactions between two adjacent bins occurs, thus resulting in a ‘strong’ insulation between TADs. A local minimum in physical interactions can still be observed between adjacent bins which results in “weak” insulated TADs.”*

Comment 3: With some exceptions (centromeric TADs and random regions along the chromosomes), TADs and boundaries do not seem to be well identified in the revised version. Several examples can be observed

in Fig1, Fig 3, Sup Fig 2, Sup Fig6 and Sup Fig7 (in most of the cases nested TADs make difficult the visualization of the identified TADs and boundaries). This could be simple due to the heterogeneity of the Hi-C signal, but also to other patterns that are observed (and not discussed). Among these patterns, we can observed inter-TADs contacts, loops, and some patters that may look like sliding loops (for instance in chromosome 8, in one of the AGRs). Furthermore, all figures displaying TADs need to be improved, they are small and their visualization is difficult. For instance, the insulation score is plotted below the contact map, but it is useless since the scale is small and the '0' is not defined. Furthermore, authors refers to 'a weak insulation score', but what does this mean? It would be useful that they define which score leads them to this conclusion: is it a value close to zero? Is it a value close to -1? What does '0' means in terms of insulation? Why the score is flat along the chromosome, except in well-defined TADs (for instance centromeric TADs). Despite good quality controls, the contacts displayed in the matrices do not correlate with the TADs and boundaries detected.

Authors: We understand the comment of the reviewer, but also feel that “eyeballing” plots is misleading. We have performed TAD predictions based on the actual calculation of insulation scores and believe that a robust prediction of nested TADs and loops is not achievable based on the current Hi-C resolution. As detailed in our response to the next comment, we have performed further validation of our TAD prediction, and of the conclusions that we obtained, and we believe that our predictions are robust (see below). Our approach, based on insulation scores, predicts a TAD boundary if a significant local minimum of physical interactions is observed between two adjacent bins. This local minimum result in a negative value when a large difference in physical interactions between two adjacent bins occurs, thus resulting in a ‘strong’ insulation between such regions. Meanwhile, local minima in physical interactions can still be observed between adjacent bins which results in “weak” insulated TADs. Finally, a value of ‘0’ means that there is no difference between two adjacent bins. Accordingly, centromeric regions show a strong difference in physical contacts within the centromere when compared with the pericentromeric region, or the two adjacent bins in terms of the insulation score. Thus, a strong local minimum of physical interaction can be observed and the variation in the insulation score is significantly high. Our TAD boundary depiction reflects such depletion of interactions as explicitly shown in Fig. 1b; Figure S1e, g-i; Figure S5d. Specifically, in Figure S1e, we measured the unbiased coverage strength (number of raw Hi-C reads). This shows that TAD boundaries detected by the insulation score calculation are depleted of Hi-C reads. We also increased the size of all the supplementary figures where a Hi-C matrix is depicted, and thus the visualization of scales is improved.

Considering that the median size of TADs in *V. dahliae* is about 100kb and the doubts raised in the method used, it would be worth to test other tools to identify TADs and boundaries (for instance: *DI* (Dixon, 2012) or *chromosight* (Matthey-Doret C, 2020)) that are known to efficiently identify smaller TADs.

Authors: There is very little information on chromatin organization and TADs in filamentous fungi, and thus it is quite impossible to say what the expected size is. As a quality control, we reasoned that an accurate TAD prediction should accurately depict centromeric TADs, considering that centromeres in *Verticillium dahliae* have been experimentally determined and verified. Thus, we compared the centromeric TAD prediction at different binning sizes to the centromeric regions defined by CenH3 and repeat content (Seidl et al., 2020), as we shown in Fig. S1g-h. The prediction at variable binning size (~4.5 kb resolution) is the closest to the orthogonal validation with the coordinates of Seidl et al., 2020. Thus, we believe that the analysis we have performed and reported is the most robust.

However, following the reviewer recommendation we now also predicted TAD boundaries using an additional tool, *chromosight* (Matthey-Doret C, 2020). As expected, we found that *chromosight* predicts smaller TADs. However, centromeric TADs were not captured as accurately as the *hicExplorer* prediction did, as can be seen in the figure below (Figure I, see below). Whereas *hicExplorer* predictions (black triangles) accurately match centromeric regions, *chromosight* (red triangles) predicts these TADs less accurate.

Identification of smaller TADs cannot only be performed with *chromosight*, but can also be captured by *hicExplorer*, by performing binning at a lower resolution. To show this, we have generated a new supplemental figure (Fig. S2). Thus, we now also cross-verified the features associated with our initial TAD prediction based on binning at 2 kb resolution that recapitulates shorter TADs and loops (Fig. S1g-h). As expected, the 539 TAD boundaries depict smaller TADs (mean = 51,529.85 bp), from these, 493 TAD boundary regions overlap with the core genome, 46 with the AGRs and five with centromeres. Importantly, at this 2 kb resolution, TAD boundary regions in the core genome and in AGRs maintain the features that we identified based on the variable binning size (~4.5 kb resolution) (Fig. S2). We observe that TADs and boundary regions differ in gene density, repeat density, chromatin accessibility, H3K27me3 and sequence conservation (Fig. S2).

Thus, in conclusion, we believe that we have performed the most robust TAD prediction based on validation based on centromeric TADs. However, even if we capture smaller TAD sizes, and thus permit centromeric TADs to be predicted sub-optimally, we maintain the previously identified

differential hallmarks of TADs and boundary regions. Thus, importantly, we show that the major conclusions of the manuscript do not immediately depend on differential TAD predictions.

We have summarized the major findings of this additional analysis in the results section of the manuscript (lines 107-124) such that this section now reads: *“Given the little information on chromatin and TAD organization in filamentous fungi, we further assessed TAD predictions at different binning thresholds (Fig. S2a), while maintaining that an accurate TAD prediction should accurately depict the previously experimentally verified centromeric TADs (Seidl et al., 2020). This analysis revealed that binning at 4 kb resolution most accurately called centromeric TADs (Fig S2c). TAD prediction based on binning at 2 kb resolution that recapitulates smaller TADs (mean size 51,5 kb) and loops, resulting in an increased number of TADs and TAD boundary regions (539) as nested TADs were identified when compared with predictions at 4 kb resolution (Fig. S2). In total, 493 TAD boundary regions overlap with the core genome, 46 with AGRs and five with centromeres. Importantly, even at this 2 kb resolution, TAD boundary regions in the core genome and in AGRs maintain the features that we identified based on the 4 kb binning resolution (Fig. S2), as we observe that TADs and boundary regions differ in gene density, repeat density, chromatin accessibility, H3K27me3 and sequence conservation (Fig. S2d). In conclusion, the most robust TAD prediction based on validation based on centromeric TADs occurs at 4 kb binning resolution. However, even if we capture smaller TAD sizes and thus reveal nested TADs, while permitting centromeric TADs to be predicted sub-optimally, we maintain the differential hallmarks of TADs and boundary regions that suggest that suggest that TADs and boundaries differ in functionality between the two genomic compartments.”*

Figure i: Distribution of TADs over all chromosomes of *Verticillium dahliae* strain JR2. From top to bottom: Chromosome sizes, Hi-C contact matrix depicting TADs predicted using *hicExplorer* (black triangles) and *chromSight* (red triangles) and centromeric regions (black bars).

Comment 7: Given the nature of the article, it is essential to display in the main text not only a contact map for the whole genome, but also a Hi-C map centered in one individual chromosome, and one showing inter-chromosome interactions (similar to Sup Fig8d).

Authors: We believe that including a contact map of the whole genome in the main text is redundant with our previous publication: Seidl et al., 2020, as this map was shown there as well. The Hi-C map centered in one individual chromosome is shown in the main Figure 1, in addition to other genomic features, like histone modifications, gene expression or repeat density. Furthermore, we generated a figure centered around chromosome 4 that shows inter-chromosomal contacts and added it to the supplementary information.

Concerning Figure S8, why the authors do not include the features of interest in S8C? In this figure it can be seen that in addition to the centromeric inter chromosome contacts, there are looped regions at TADs boundaries. What are these contacts? Are they ARGs?

Authors: Supplementary figure 8 depicts the matrix interaction of the eight chromosomes of *Verticillium dahliae* JR2. The vast majority of the inter-chromosomal interactions are centromeric interactions. Such interactions were previously shown in Seidl et al., 2020. At this resolution is very difficult to disentangle TAD boundaries.

Why Figure S8C shows almost no contacts? For this figure, it would be nice to show the map for the whole chromosome and a magnification of the contacts so that the reader can have the 3D context of the interactions.

Authors: As expected, the scale and intensity of the inter-chromosomal interactions is much smaller when compared with the local interactions. We added an additional figure to panel S8D to describe the context of inter-chromosomal interactions colocalization with duplicated regions. Together with the additional figure displaying local and inter-chromosomal interactions of chromosome 4, we now provide much more context for the reader.

Other Important points:

In the abstract, the authors stated: 'Intriguingly, TADs are less clearly structured in AGRs than in the core genome...' This is a strong confirmation that needs to be shown. The authors mentioned that the insulation score is weaker in AGRs than in other areas, how this is connected to a less structured TAD? . If authors wish to state this, it would be a good idea to analyze contact dynamics within ARGs and core genome, and defined after this what it is 'a clearly structured region'.

Authors: We now rephrased the sentence to “Intriguingly, TADs are less clearly insulated in AGRs than in the core genome.....”.

Circular plots are helpful (probably better placed in supplementary figure) but undoubtedly, matrices showing ARGs inter-contacts need to be presented, given that this is a major finding of the article.

Authors: We believe that circular plots are a more intuitive visualization for a broader audience considering the major findings we want to show: the physical interaction of AGR regions within the nucleus. However, in the current version of the manuscript, we include the respective matrices of such physical interactions in the Supplementary Figure 9.

Reviewer #5 (Remarks to the Author):

The revised version of the article 'Implications of the three-dimensional chromatin organization for genome evolution in a fungal plant pathogen' by Torres and collaborators addresses the major points raised, particularly those related to TADs and TAD detection. The new supplementary figures enhance the clarity of the article.